# New Frontiers about the Role of Human Microbiota in Immunotherapy: The Immune Checkpoint Inhibitors and CAR T-Cell Therapy Era

**DOI:** 10.3390/ijms21238902

**Published:** 2020-11-24

**Authors:** Vanessa Innao, Andrea Gaetano Allegra, Caterina Musolino, Alessandro Allegra

**Affiliations:** 1Division of Hematology, Department of Human Pathology in Adulthood and Childhood, University of Messina, 98122 Messina, Italy; cmusolino@unime.it; 2Radiation Oncology Unit, Department of Biomedical, Experimental, and Clinical Sciences “Mario Serio”, Azienda Ospedaliero-Universitaria Careggi, University of Florence, 50100 Florence, Italy; andrea.allegra@hotmail.it

**Keywords:** microbiota, Immune Checkpoint Inhibitors, CAR T-cells, probiotics, cancer

## Abstract

Microbiota is considered an independent organ with the capability to modulate tumor growth and response to therapies. In the chemo-free era, the use of new immunotherapies, more selective and effective and less toxic, led to the extension of overall survival of patients, subject to their ability to not stop treatment. This has focused scientists’ attention to optimize responses by understanding and changing microbiota composition. While we have obtained abundant data from studies in oncologic and hematologic patients receiving conventional chemotherapy, we have less data about alterations in intestinal flora in those undergoing immunotherapy, especially based on Chimeric Antigen Receptor (CAR) T-cells. Actually, we know that the efficacy of Programmed Cell Death 1 (PD-1), PD-1 ligand, and Cytotoxic T lymphocyte-associated protein 4 (CTLA-4) is improved by probiotics rich in *Bifidobacterium* spp., while compounds of *Bacteroidales* and *Burkholderiales* protect from the development of the anti-CTLA-4-induced colitis in mouse models. CAR T-cell therapy seems to not be interfering with microbiota; however, the numerous previous therapies may have caused permanent damage, thus obscuring the data we might have obtained. Therefore, this review opens a new chapter to transfer known acquisitions to a typology of patients destined to grow.

## 1. Introduction: The New Era of Immunotherapy

Nowadays, immunotherapy is considered the latest frontier in the fight against cancer, relying on revolutionary concepts to treat tumors like an infectious disease. Several lines of evidence relate the state of health to the gut microbial composition, especially in the case of autoimmune diseases (e.g., type 1 diabetes mellitus, Hashimoto thyroiditis, rheumatoid arthritis, inflammatory bowel diseases, etc.). A series of scientific reports indicate the possibility to use the intestinal microbiome as a marker capable of predicting the response to immunotherapy treatment, which is better in patients with a very rich of different species of intestinal microbiome.

Old acquisitions reported that cancer cells proliferation is a consequence of failure of endogenous immunological control [1], through several circumvention mechanisms to escape the host’s immune system action, such as the downregulation of target antigens or creating a cellular microenvironment with immunosuppressive characteristics. This is mostly exploding as an acquired immune-paresis of T-cells: Physiologically, when they recognize an antigen on a tumor cell, this is rapidly killed by activated CD8+ cytotoxic T cells, helped by CD4+ T helper cell type 1 (Th1). Really, over the years, several models for the immune response against cancers have been proposed, including the old most reliable concept of immunosurveillance and the danger model. The first explains the immune response as a reaction to the recognition as “non-self” of antigens expressed by cancer cells [2], while the second is more useful to understand the mechanisms for circumventing this control. According to the danger model, in fact, the antigen-presenting cells (APC), such as dendritic cells, activated macrophages, and B-cells, play a key role in detecting the danger signals of tumor cells, stimulating and amplifying the T cell response. Therefore, the ineffectiveness of this response would be secondary to the inability to recognize tumor cells as “dangerous”, or to their ability to “disguise” themselves by non-dangerous cells [3]. More recently, the “missing self” hypothesis supported the important role of natural killer (NK) cells in immune surveillance: Controlled by a delicate balance of positive and negative signals, they are able to kill cells missing of Mayor Histocompatibility Complex (MHC) class I [4]. About that, we know that NK cells are more represented in tumor tissues and peripheral blood of patients with Multiple Myeloma (MM) than in healthy controls, observing a significant increase of CD94lowCD56dim NK cell subset, also present in its preclinical conditions, known as monoclonal gammopathy of undetermined significance (MGUS) and smoldering MM. Analyzing their killing abilities, we demonstrated that they represent the main cytotoxic NK cell-type against MM cells that are themselves able to determine their rapid expansion [5].

In this scenario, the use of a combination treatment capable to enhance the anticancer response of the same immune system cells, or even better the ability to re-educate the same cells of the patient’s immune system to respond to the uncontrolled tumor cells growth represent the keystone for the future of the cancer therapy. Known as Immunological Checkpoint Inhibitors (ICIs), these are compounds with therapeutic action played through control of the intensity and duration of the immune response against several types of cancer. Simplistically, these substances “inhibit the inhibition” of immune system, so inducing tumor cells apoptosis.

We know that ICIs are effective in approximately half of patients with metastatic melanoma, they may lead to serious side effects, and the duration of the response to treatment may be limited. In recent years a series of evidences indicates a role of the intestinal microbiome in influencing the success of immunotherapy. About that, the use of antibiotics and some probiotics may reduce the effectiveness of treatment, while, on the contrary, some bacterial strains seem to increase the effectiveness of treatment. Manipulating the microbiome might be a way to overcome the problem of resistance to antitumor treatments.

In autoimmune disease, Programmed Cell Death 1 (PD-1), PD-1 ligand (PD-L1), and Cytotoxic T-lymphocyte-associated protein 4 (CTLA-4) are now well recognized as co-inhibitory molecules able to brake and modulate the immune response. Co-inhibitory ligands and receptors are often overexpressed in the cancer cells, stromal cells, and tumor microenvironment, also playing a central role in mechanisms of immune tolerance. Specifically, when binds its ligand PD-L1, PD1 can inhibit the immune activity of T-cells issuing co-inhibitory signals, thus causing tumor cells escape [6]. The FDA already approved specific monoclonal antibodies, known as ICIs, active against PD-1 (nivolumab), PD-L1 (pembrolizumab), and CTLA-4 (ipilimumab) for several types of tumor, such as melanoma, kidney, bladder and lung cancers, and Hodgkin lymphoma. In these diseases, they have been proven effective in reactivating the host’s endogenous anticancer immune response, but with high inter-individual heterogeneity [7]. Although the mechanisms underlying this inter-individual variability in the response to ICIs therapies are not yet fully clear, it would appear that a patient’s gut microbiota plays a central key role.

More recently, the strategy of genetically engineering T lymphocytes to become active against cancer cells, has determined the birth of CAR-T Therapy (Chimeric Antigen Receptor T-cell therapies). Born in the 1990s and used as therapy in 2011, this idea was set up by the Nobel Prize in 2018.

Specifically, T cells are taken from the patient’s blood, and then they are genetically modified to express CAR receptor on their surface, so that you can increase the immune response, and re-infused in the patient. Unlike the checkpoint inhibitors strategy, CAR-T represents a “custom cancer medicine”. Started on patient’s immune cells, each dose is developed and produced for every single one of them. The first approval for the application of CAR-T in patients with certain blood cancers (Acute Lymphoblastic Leukemia and Non-Hodgkin Lymphoma in adult) arrived in 2017 in the United States and in 2018 in Europe, extending today their applicability also to patients with Multiple Myeloma and Hodgkin Lymphoma in onco-hematological field, by also laying the foundations for the development of “Senolytic CAR T-cell therapy” in age-related pathologies, such as diabetes, atherosclerosis, osteoarthritis, and fibrosis. Unable to die via apoptosis, senescent cells, also called “zombie cells”, are “immortalized” in a stable cell-cycle arrest. On the one hand, they have an important physiological role in cancer suppression, preventing the expansion of pre-malignant cells, and they also have a beneficial role in responding to wound healing; on the other hand, they produce inflammatory cytokines that lead to diffuse chronic tissue damage, by determining typical alterations of the older age. About that, a very recent study reports the results of specific CAR T-cells able to target antigens produced by senescent cells in vitro and in vivo, such as the urokinase-type plasminogen activator receptor (uPAR), named uPAR-specific CAR T-cells. In mice with lung adenocarcinoma, previously treated with senescence-inducing combination of drugs, and in mice with chemical or diet-induced liver fibrosis, the uPAR-specific CAR T-cells demonstrated to prolong survival, by proposing their therapeutic potential for senescence-associated disease [8]. Similarly, others specific anti-inflammatory CAR T-cells are being studied to treat myasthenia gravis (MG) and systemic lupus erythematosus (SLE). The action of CAR T-cells is addressed against plasma cells, expressing both BCMA and CD19, respectively, responsible for the pathogenesis for neuronal damage in MG and multifocal damage in SLE [9,10]. Finally, using the genome editing technique CRISPR (Clustered Regularly Interspaced Short Palindromic Repeats), some scientists from Seattle Children’s Research Institute and Benaroya Research Institute transformed a subtype of CD4 T cells, known as Thymic regulatory T cells (tT_regs_) into cells with immunosuppressive properties, with the aim of controlling autoimmune response in type 1 diabetes. Present in very small quantities in peripheral blood and physiologically characterized by powerful inhibitor action on autoreactive immune responses, the tT_regs_ have a powerful function to prevent the activation of autoimmune response. Using CRISPR-Cas9 system, these scientists created autologous T_reg_-like cells amplifying the expression of the master transcription factor for tT_regs_ FOXP3, inserting a powerful enhancer/promoter to its first coding exon, thus leading immunosuppression in vivo human and murine models of the disease [11].

Unfortunately, only 40% of patients treated with CAR T-cells achieve a response, and the high percentage of failures has led scientists to investigate the possible reasons for this, finding out that one of the resistance mechanisms depends on the patient’s microbiome.

In the era of chemo-free therapy, the importance of environmental factors impacting on the functions of the immune system has also been reassessed. Among these, the diet represents a key aspect for the evolution of all living beings, especially in patients affected with chronic disease such as tumor, although the role of specific dietary changes in altering the immune system and contributing to the response of cancer cells to immunotherapy remains unclear. A number of studies are being developed in this direction, and one of the most important topics of attention is intestinal microbiota.

## 2. The Close Correlation between Microbiota and Tumors

The balance between somatic cells and intestinal microbiome is the result of the interaction of the latter with the immune system, reflecting itself in the formation and progression of tumors and not just those of the intestines. For example, studies in colorectal cancer have shown that the microbiome of cancer patients is very different from that of healthy subjects [12]. Mechanisms are not fully known but include the production of metabolites and toxins and modulation of the immune system. Furthermore, several studies suggest a causal role of the microbiome, particularly the intestinal one, in determining the effectiveness to conventional cancer therapies and immunotherapy, as well as modulating responses and also the susceptibility to side effects of these therapies [13,14].

However, what is microbiota, and how does it work? The microbiota is a very large (trillion) set of micro-organisms, such as viruses, mushrooms, protozoa, and especially bacteria, which populate various body surfaces, mainly intestinal ones. It has a weight of about 1 kg and is therefore considered the fourth organ of the digestive tract. It is characterized by a huge diversity, with over 1000 bacterial species detected in the entire human population, about 150 of which are present in the individual. The microbiome, which is the gene complex (genome) of the intestinal microbiota, contains at least 3.3 million microbial genes, at least 150 times the human genome, which allows us to say that we are, in a way, the guests of the microbiota and not the other way around [15]. The symbiotic relationship between host and microbiota represents the key to maintaining the host’s health status. In order to do this, the intestinal microbiota needs many species, and the most bacterial types are *Firmicutes* and *Bacteroides*, followed by *Actinobacteria* and *Proteobacteria*. Their functions are multiple: metabolic (amino acids synthesis, absorption of fat-soluble vitamins, production of short-chain fatty acids, and intervention in the metabolic processes of proteins, lipids, and carbohydrates), protective (prevention of colonization by pathogens and regulation of the innate and adaptive immune system), and structural (regulation of intestinal barrier permeability). Microbiome can be modulated in various ways, such as taking prebiotic fibers (to stimulate the growth of preexisting bacteria) and probiotics (live bacteria capable to repopulate flora and improve its variability), by doing physical activity (it has been shown that those who practice at least 150 to 180 min per week of physical activity have better microbiota and a more efficient immune system than sedentary people). On the contrary, some classes of drugs (antibiotics, proton pump inhibitors, and certain antidepressants), cigarette smoke, and chronic stress depress bacterial variability with negative effects on the immune system. Moreover, microbiota fecal transplant, today successfully used for the treatment of *Clostridium difficile* diarrhea, can change microbiome, and again, breast milk intake in the first six months of life and the diet throughout life.

Diet plays a decisive role in the intestinal microbiota, especially about dietary fibers. These, in fact, arrive undigested in the colon and undergo a fermentation process by intestinal bacteria, finally producing metabolites, such as short-chain fatty acids, including butyric acid, propionic acid, and acetic acid. In addition, to reduce the colic pH with protective function against pathogenic bacteria, these metabolites perform a nourishing activity for intestinal epithelial cells, strengthening tight-junction, reducing leaky gut, and establishing an anti-inflammatory environment. Finally, the production of anti-inflammatory cytokines, such as IL-10 and IL-22, can be stimulated by some molecules contained in foods, like the antioxidant catechins contained in the green tea, the quercetin of wild berries, curcuma, vitamins A and D, vitamin E of extra-virgin olive oil, the resveratrol of red wine, and the fish omega-3.

In recent years, many studies focused about changes in the microbiota caused by Mediterranean, oriental, vegan, and gluten-free diets. Good bacterial species, like *Bifidobacterium* and *Eubacterium* spp., are reduced by a diet poor in fiber, but high in animal fat and proteins [16].

It is well-known that microbiota directly stimulates local intestinal immunity, increasing toll-like receptor (TLR) expression, antibody secretion, and CD4+ T-cells production. The lipopolysaccharide produced by microbial species can upregulate TLRs, thus provoking nuclear factor-kB (NF-kB) activation, and then controlling cancer cells survival, growth, invasion, and tumor-associated inflammation [17,18]. In addition, T-helper cells (Th17) play an important role in tumorigenesis, especially when the balance Th17/T_regs_ is altered, and it is demonstrated that *Bacteroides fragilis* induces a Th17 response in animal [19,20]. Furthermore, segmented filamentous bacteria increase IL-10, IL-17, and IFN-g production, the increase of which is also due to the presence of human commensal bacteria, such as *Bifidobacterium longum* and *Bacteroides thetaiotaomicron*, responsible of TNF-a pathway activation too [21]. Moreover, oxidative stress plays a major role in induction of tumor expansion. Several food metabolites have been shown to be responsible for amplifying epigenetic damage, promoting the instability of DNA. It is known that co-metabolism of xenobiotic, such as azoxymethane and irinotecan, produces significant quantities carcinogenic factors, by bacterial β-glucuronidases [22]. Similarly, microbial catabolism of proteins produces several N-nitroso elements (NOCs) responsible for DNA alkylation. Moreover, metabolism of aromatic amino acids results in the production of indoles, phenols, phenylacetic acid, p-cresol, and polyamines, whose catabolism creates an oxidative environment and then tumors [23]. NOCs are, in fact, considered the most potent pancreatic carcinogens, by N-nitrosodiethylamine (NDEA) and N-nitrosodimethylamine (NDMA), their most prevalent compounds contained in foods [24].

## 3. Intestinal Microbiota Impacts on Conventional Chemotherapy in Hematologic and Oncologic Neoplasms

Recently, several preclinical and clinical reports about some type of tumor supported the importance of intestinal microbiota in the modulation of host response to chemotherapy, immunotherapy, radiotherapy, and even surgery [25,26]. The immunosuppressive action of conventional chemotherapy, responsible for hematological and gastrointestinal toxicities, also severe, has limited the study of the positive impact of the microbiota on the host’s anticancer response [27]. In fact, the direct damage caused by cytotoxic drugs on intestinal cells is undoubted (just think about direct action on bowel villi and mucosal damage by Doxorubicin and Cyclophosphamide), and the antibiotic prophylaxis or therapy further worsen the alteration of the intestinal microbiota [28,29,30].

To better realize these biological mechanisms, the framework known as “TIMER” has made its way: Translocation, Immunomodulation, Metabolism, Enzymatic degradation, and Reduced diversity and ecological variation represent the building blocks of pharmacomicrobiomics in the optimization of anticancer therapy today [26,31].

Several authors reported changes in response to chemotherapy, and then its prognostic impact on overall survival of the patients, related to the microbiota damage. Moreover, the metabolization of cytotoxic drugs by bacteria influences the effectiveness of chemotherapy. In this regard, investigating 30 antitumoral agents, in 2015, an intergroup of Ireland and Canadian researchers showed a direct inhibition of 10 different drugs (Cladribrine, Vidarabine, Gemcitabine, Doxorubicin, Daunorubicin, Idarubicin, Etoposide phosphilate, Mitoxantrone, B-Laoachine, and Menadione), by certain bacterial species (Gram-negative non-pathogenic *E. coli* and Gram-positive *Listeria welshimeri*), while in general the same bacteria increased the efficacies of six others (5-fluorocytosine, 6-Mercaptopurine-2′-deoxyriboside, AQ4N, and CB1954), due to direct changes to their chemical structure [32]. Similarly, Cyclophosphamide (CTX) and platinum compounds, massively used against hematologic disease and oncologic neoplasms, cause a direct damage of epithelial cells of the small intestine. Thus, CTX caused migration of lots of bacteria into lymphatic organs, responsible of an increased antitumoral response by T-helper cells (Th17), and an improvement of therapeutic effect of drug too [33]. Differently, platin treatment was less effective in germ-free mice previously treated with broad-spectrum antibiotics (such as vancomycin and colistin) [34], while *Lactobacillus acidophilus* increased its anticancer action, and also the effectiveness of immunotherapy in murine colon cancer models [26,35]. However, this reality is upside down in the case of therapy with some Immune Checkpoints Inhibitors, such as anti-CTLA-4, the effectiveness of which has improved by the concomitant use of vancomycin, that preserves the Gram-negative species, such as *Burkholderiales* and *Bacteroidales*, decreasing Gram-positive intestinal bacteria at the same time [36].

High doses chemotherapies used in hematological disease as conditioning for auto- or allo-HSCT frequently cause bacteremia, often vancomycin-resistant [37], due to gut expansion of *Enterococcus, Streptococcus,* and *Proteobacteria* spp., thus indicating these bacterial species as predictive markers of bacteremia before and during chemotherapy with different risk profiles [38]. In this regard, in the same group of patients, a gut microbiota rich in *Barnesiella* spp. presented low-risk profile to develop bacteremia, through direct inhibition of intestinal colonization by vancomycin-resistant *Enterococcus faecium* (VRE) [39].

In 2017, an English report pointed out the importance of microbiota to modulate the host response to chemotherapeutic drugs, sustaining its role in facilitate drug efficacy, abrogate drugs’ anticancer effects, and mediate their toxicity. Thus, accepting this assumption, they proposed the idea to develop personalized anticancer strategies of therapy, implementing a better knowledge of the co-metabolism of drugs by intestinal bacterial species. This concept is not demonstrable only for conventional chemotherapy, but also for the novel targeted immunotherapies, such as anti-PD-L1 and anti-CLTA-4 therapies. The negative side of the medal is represented by the cases of lethality due to increased toxicity of chemotherapy drugs caused by their xenometabolism; for instance, several years ago, Japanese authors reported the accumulation in blood of 5-fluorouracil (5-FU) sorivudine bi-therapy metabolites caused by *Bacteroides* spp, [40,41]. The same 5-FU, together with doxorubicin and irinotecan, is responsible for increasing *Staphylococcus* and *Clostridium* spp., and decreasing Enterobacteriaceae, *Lactobacillus*, and *Bacteroides*, thus leading to intestinal and oral dysbiosis, resultant in typical mucositis [42,43]. Similarly, like *Bacteroides* spp., all b-glucuronidase-producing bacteria, such as *Clostridium* spp. or *Faecalibacterium prausnitzii,* caused a toxic increase of irinotecan active metabolite SN-38 in the gut of patients with colorectal cancer, resulting in diarrhea [44,45] (Figure 1).

In patients with neuroendocrine tumors of the midgut, an increase of the same *Faecalibacterium prausnitzii* was observed following chemotherapy [46].

Furthermore, the alterations of microbiota were held accountable in oxaliplatin (OXA) chemoresistance in colorectal cancer and lymphoma, without understanding of the specific changes that were responsible for it, as observed in treatments with CTX, for which efficacy is directly correlated with the presence of *Enterococcus hirae* and *Barnesiella intestinihominis*, probably via reactive oxygen species (ROS) production [47]. Some intratumoral bacteria have demonstrated the ability to change the effectiveness of chemotherapy through active metabolism of them (Figure 1). In this regard, a decrease of antitumoral efficacy of pyrimidine nucleoside analogues, such as gemcitabine (GTB), is reported in *Mycoplasma hominis* infected cell lines and in human pancreatic adenocarcinoma, due to the direct degradation of these drugs in the tumor cells by mycoplasma thymidine phosphorylase, ever rehabilitated by specific inhibitors of this enzyme [48,49]. Moreover, in colorectal cell lines and colon cancer patients the inducted autophagy by *Fusobacterium nucleatum* can be held accountable for the resistance to OXA and 5-FU [50,51].

Among patients diagnosed with metastatic renal cell carcinoma, some authors identified two different risk groups to develop diarrhea following Vascular Endothelial Growth Factor-Tyrosine Kinase Inhibitors (VEGF-TKI) treatment: the low-risk group, with an higher concentration of *Prevotella* and low levels of *Bacteroides*, and the high-risk one, with an opposite condition, suggesting a close relationship between microbiota and drugs’ toxicity [52].

It has also been seen that a wide heterogeneity of intestinal bacterial species at diagnosis protects subjects with acute myeloid leukemia (AML) from infectious complications after induction chemotherapy [53], independently of age, chemotherapy regime, or type and duration of antibiotic therapy used; differently, an increased amount of *Stenotrophomonas* spp. seems to be responsible of higher risk of infection in the same cohort [54]. Similar results were showed in patients with non-Hodgkin disease, who developed bloodstream infections, probably correlated to the minor diversity in gut microbiota observed in their fecal samples even before to start of chemotherapy, in which the reduction of *Barvesiellaceae, Christensenellaceae,* and *Faecalibacterium* was mainly observed [55].

These assumptions led the studies about fecal microbiota transplantation (FMT), a compelling tool to restore healthy intestinal environment; however, currently, its role in transplant recipients is not well established [56].

## 4. Microbiota and Immunotherapy in Oncologic and Hematologic Neoplasms

The studies previously reported are only a few among more interesting works about the role of gut microbiota into modulation of host response and effectiveness of conventional chemotherapy agents, but the aim of our reports is to investigate the changes in microbiota with next novel chemo-free anticancer treatment, so to try useful tools to optimize this response by modifying the feeding of each individual patient, based on types of drugs and cancer.

Several evidences relate to the state of health to the gut microbial composition, especially in the case of autoimmune diseases (e.g., type 1 diabetes mellitus, Hashimoto thyroiditis, rheumatoid arthritis, inflammatory bowel diseases, etc.), but not limited to them. A series of scientific reports indicates the possibility to use intestinal microbiome as a marker capable of predicting the response to immunotherapy treatment, better in patients with a very rich of different species intestinal microbiome.

Actually, in oncologic cancers ICIs are not the future anymore, but they tell the contemporary history of the fight against cancer. Several authors have described the close reciprocity of the interaction between the composition of the microbiota and the response to immunotherapy.

In renal cell carcinoma, non-small cell lung cancer, and unresectable or metastatic melanoma, anti-PD1 or anti-CTLA-4 are routinely used, with good responses, and with major evidence that they are often influenced by particular gut microbial sets [57,58,59,60,61,62].

We know that ICIs are effective in approximately half of patients with metastatic melanoma, may lead to serious side effects, and the duration of the response to treatment may be limited. In recent years, a series of evidences indicate a role of the intestinal microbiome in influencing the success or not of immunotherapy. About that, the use of antibiotics and some probiotics may reduce the effectiveness of treatment; on the contrary, some bacterial strains seem to increase the effectiveness of treatment. Manipulating the microbiome might be a way to overcome the problem of resistance to antitumor treatments.

In this respect, some authors have identified a number of bacterial strains that could enhance the immune antitumor response and some biomarkers of response to ICIs. A report of more than 40 scientists coordinated by the Sanford Burnham Preby Medical Discovery Institute demonstrated a causal relationship between intestinal microbiome and the ability of the immune system to fight cancer. In particular, the authors identified a cocktail of 11 bacterial strains capable of “activating” the immune system and slowing down the growth of melanoma in mice. The study also highlighted the role of a cell signal pathway activated in stress response, UPR (unfolded protein responded), implicated in protein homeostasis. In particular, a reduced URP has been observed in patients with melanoma responding to immunotherapy, and this may represent a marker potential for stratification of responders and non-responders [63]. One of the models used is mice without the RNF5 gene (RING finger protein 5), a ubiquitino-ligase removing damaged or poor folded proteins. These animals were able to inhibit the growth of melanoma in the presence of an intact immune system and an intact intestinal microbiome, while lost their anticancer immune phenotype, if treated with a cocktail of antibiotics or if placed in the same cage with normal mice. This led researchers to show that a reduction of UPR in immune cells and intestinal epithelium cells causes the activation of immune cells. Then, researchers analyzed tissue samples from three cohorts of patients with metastatic melanoma treated with checkpoint inhibitors. A reduced expression of various components of UPR (sXBP1, ATF4, and BiP) was related to the response to treatment, suggesting their use as response biomarkers to treatment with ICIs, so that they could select patients to be treated with these compounds. The next steps of research will be to determine which bacterial metabolites can reduce the tumor growth, also suggesting if the use of particular prebiotics could increase favorable intestinal bacterial populations.

Therefore, can intestinal microbiome help to improve the response to immunotherapy used in the treatment of several types of tumors? The answer is yes. There are favorable microbiomes and adverse microbiomes: Patients who responded to therapy had a different microbiome, because of the composition and wealth of species, compared to those of patients who had not responded. Thanks to sophisticated techniques, researchers identified the species prevailing in patients who responded to therapy.

In this regard, several studies demonstrated that some bacterial species, such as *Bifidobacterium, A. muciniphila*, and *Faecalibacterium*, improve the efficacy of nivolumab, that was reduced in patients who received antibiotic therapy before or soon after ICIs treatment. Changes in microbiota also occur following administration of anti-CTLA-4 immunotherapy, characterized by a rapid increase of *Bacteroidales* spp. and a decrease of *Burkholderiales* [36] (Figure 2). In addition, this report showed that the effectiveness of anti-CTLA-4 immunotherapy was reduced in germ-free mice, and reprised after oral assumption of *Bacteroides thetaiotaomicron* or *Bacteroides fragilis*; besides this, an increase of *Bacteroides* spp. reduced ipilimumab-induced colitis and diarrhea [57], probably due to accumulation and maturation of plasmacytoid dendritic cells in mesenteric lymph nodes, finally responsible for stimulating Treg-cell proliferation in the lamina propria [64]. On the contrary, they also observed that a reduction or a lack of cellular polyamine transport systems and B vitamin synthesis correlated with increased risk of ipilimumab-induced colitis.

About that, a few months ago, the prestigious *Science* magazine published three studies of independent groups, which prove that resistance to ICIs is to be attributed to an abnormal composition of the intestinal microbiota [65,66,67]. The presence of microbiota is necessary for the response to ICI, because the benefits of treatment were reduced in patients who had taken antibiotics. The T-lymphocyte response competence is complementary to gut microbiota, and patients with active T-cell response in tumor microenvironment achieve better response to PD-1 inhibitors [68].

Using the fecal microbiota transplant (FMT) technique, the study of Sivan et al. is particularly well-known. It reported different anti-PD-L1 response rates on wall models with different contents of the gut microbiota, reversely related to tumor growth rates. The subsequent FMT from the first to the second species modified the response [69]. By using 16S ribosomal RNA gene sequences, they showed an increase of antitumor T-cell activation in better responders, who also had an increased representation of *Bifidobacterium* spp. They confirmed the upregulation of tumor-specific T-cells activity in the tumor microenvironment, followed by an improved control of tumor growth, by treating mice with commercial probiotics rich in *Bifidobacterium*, whose beneficial effects were invalidated by their heat inactivation and by depletion of CD8+ T cells. Finally, *Bifidobacterium* spp. Appear to be able to communicate with dendritic cells in the enhancing action of T cells.

In particular, Zitvogel’s team assessed the response to PD-1 inhibitors and their ligands PD-L1. Despite the fact that these monoclonal antibodies are the most used ICIs in current therapy, clinical benefits are only evident in around a quarter of the patients treated. Analyzing 249 patients with different types of cancer (melanoma, lung, kidney, and bladder) treated with PD-1/PDL-1 inhibitors, 69 of whom have taken antibiotics for non-cancer-related reasons within a time period close to the study, they reported that patients who had antibiotics previously have shown a general reduction in therapeutic response to PD-1/PDL-1 and consequently a minor PFS and OS, thus confirming how intestinal dysbiosis can affect the clinical efficacy of ICIs [36]. A comparative analysis of microbiomes obtained with fecal samples of responder (R) and non-responder (NR) patients showed many differences in the expression of the *Akkermansia muciniphila* spp., which is most likely to be associated with an increase of more than three months in PFS. In this group, *Ruminococcus* spp., *Alistipes* spp., and *Eubacterium* spp. were overexpressed, while less quantities of *Bifidobacterium adolescentis, Bifidobacterium longum*, and *Parabacteroides distasonis* were detected, as compared to NR. Finally, to test the effective correlation between *Akkermansia muciniphila* and response to PD-1/PDL-1 inhibitors, an oral transposition of fecal microbiome of R and NR patients on germ-free murine models with no intestinal microbiome was made, demonstrating best responses to ICIs in mice that had received fecal microbiome from respondents, so with marked presence of *Akkermansia muciniphila.* In addition, a significant reduction in the tumor dimension and increased immune cell accumulation at the cancerous microenvironment level were reported. It would appear to increase the release of cytokine IL-12, which supports the role of T-lymphocytes in response to the significant presence of *Akkermansia muciniphila*. However, the mechanism of action with which these bacterial species can influence the response to immune-cancer drugs remains unclear. Furthermore, if the techniques about the study of microbiome have made great progress, still, today, our ability to change it is based on diet, though not fully understanding which foods can effectively change its composition. To overcome this issue, the fecal transplant offers important advantages: It can modify patients’ microbiome without determining its real composition.

A study on MCA205 sarcoma cell lines reported that specific monoclonal antibodies anti-CTLA-4 controlled tumor progression, but the facilitation of the immunological response was expanded specifically in specific pathogen-free mice, not in germ-free ones or in others treated with broad-spectrum antibiotics, due to decreased activation of splenic CD4+ T cells and tumor-infiltrating lymphocytes [36].

## 5. Microbiota and CAR T-Cells Therapy in Hematologic Neoplasms

Adoptive cell therapy represents the latest novel and promising approach in the treatment of various types of hematological neoplasms, more than the stimulation of the T-cell response by vaccines [70]. The short time since clinical approval of CAR T-cells therapy justifies the absence of studies about interaction and effects of this therapy in relation to microbiota changes. Less than a year ago, the FDA approved CAR T-cells for treatment of relapsed/refractory patients diagnosed with Diffuse Large B-Cell Lymphoma (DLBCL) and in adult and children Acute Lymphoblastic Leukemia patients [71]. The technique provides to express for a particular chimeric receptor targeted against a tumor antigen by patients’ engineering lymphocytes previously collected. Thus, acting as antigen-presenting cells, the T-lymphocytes so modified cause the destruction of the tumor cells through the activation of the immune system. The near future will offer the same compounds but realized with allogeneic T-cells from healthy donors, tandem CAR T-cells with chimeric receptors for two ligand-binding domains, multi-CAR T-cells for different tumor antigens, complexed built-in-CAR T-cells tie up with anti-PD-L1 moAbs later released into the tumor [72,73,74]. The last category of CAR T-cells allows us to be hopeful that these compounds will improve their effectiveness even in non-hematological malignancies, on which they have not shown the expected results. That’s probably explained by the cellular antigens’ heterogeneity and the absence of tumoral specific antigens [75]. In this way, autologous tumoral-infiltrating lymphocytes (TIL) and autologous/engineered NK cells following patient’s lymphodepletion offer promising results when infused in some cancer types [76,77].

Despite the numerous data to support the role of the microbiota in changing the effectiveness and toxicity of chemotherapy, radiotherapy, and even treatment with ICIs, there are no similar studies of CAR T-cells therapy to date. We know that long-term broad-spectrum antibiotic exposure does not reduce the efficacy of CD19-targeting CAR T-cells in B-cell Lymphoma [78].

The absence of neutropenia caused by these compounds is likely to be the basis of this lack of interrelationship. Furthermore, information obtained on patients accessing this type of medicinal product may be altered by the many previously received chemotherapy lines. These are heavily pretreated patients with likely high rates of toxicities resulting from conventional chemotherapy, which could also be expected to be continued months after the end of treatment.

Finally, for both axicabtagene ciloleucel and tisagenlecleucel, no gastrointestinal adverse events have been reported, nor deep mucositis, so there is reason to believe that they are the most distant drugs from altering the intestinal microbiota [79,80].

Similarly, belantamab mafodotin (GSK2857916), a human moAb against B-Cell Maturation Antigen (BCMA) conjugated with cytotoxic antigen monomethyl auristatin F (MMAF), is the present of CAR T-cell therapy in relapsed/refractory Multiple Myeloma patients, who received previously at least four therapies. As a new therapy, here again there are no data to support the relationship with the intestinal microbiota, nor to confirm the absence of direct intestinal toxicity reported [81].

Given the short time taken by their approval, there are no data about the relationship with microbiota in terms of efficacy and tolerability to date. Therefore, controlled studies are required to analyze the fecal bacterial populations in this type of patients. Finally, to the wide variability of the sample due to the abundance of previous therapies makes it difficult to achieve unique results.

However, broad spectrum antibiotic prophylaxis for patients undergoing CAR T-cells therapy is recommended, as well as those receiving autologous transplantation [82].

## 6. Probiotics in Clinical Practice

In conclusion, we can assume that studies on the usefulness of probiotics as adjuvants to immunotherapy are still in place. In this respect, the results of many international clinical trials on large patient populations are expected, such as the “Gut Microbiome and Gastrointestinal Toxicities as Determinates of Response to Neoadiuvant Chemo for Advanced Breast Cancer” (NCT02696759), promoting by the University of Arkansas, the Chinese “Intestinal Microflora in Lung Cancer After Chemotherapy” (NCT02771470), or the sponsored “Prevention of Febrile Neutropenia by Synbiotics in Pediatric Cancer Patients (FENSY)” (NCT02544685). The latest, particularly, focus the interest on the importance of one of the most common life-threatening treatment-related complications for patients receiving intensive chemotherapy, responsible for infection due to endogenous flora alterations. In this regard, probiotics seems to be not only capable of preserving “good” gut microbiota but may also reduce the very duration of neutropenia.

The importance of selecting or increasing the presence of certain intestinal bacterial species is not exclusively in direct interaction with specific drugs, but also in indirect improvement of their effectiveness, reducing their intestinal side effects, such as mucositis, diarrhea, and bacteremia, which are often causes of discontinuation of therapies. Gastrointestinal mucositis occurs in about 50% of all cancer patients [83,84].

Patients receiving cytotoxic therapy exhibit most frequently decreased levels of *Bifidobacterium, Clostridium cluster XIVa*, and *Faecalibacterium prausnitzii*, and increased levels of *Enterobacteriaceae* and *Bacteroides*. A “good” microbiota to prevent gemcitabine-induced mucositis contains a higher concentration of Prevotella and low levels of *Bacteroides* [52]. Supplementation of *Lactobacillus rhamnosus* during 5-FU treatment in colorectal cancer patients decreases the incidence of grade three and four diarrhea and dyspepsia [85]. Similarly, in pediatric patients undergoing chemotherapy for various cancer probiotics rich in *Bifidobacterium breve* reduce incidence of fever, by prevention of the increase of bad *Enterobacteriaceae* [86]. In patients undergoing to auto/allo-HSCT probiotics containing *Barnesiella* spp. reduce the risk of developing bacteremia, causing inhibition of intestinal colonization by vancomycin-resistant *Enterococcus faecium* (VRE) [39].

Pending the results of these large trials, we can actually diversify the feeding and type of probiotics to use by type of tumor and treatment regimen performed. In fact, we know that compounds of *Bacteroidales* and *Burkholderiales* protect from the development of the ipilimumab-induced colitis in mouse models. Furthermore, it is well-known that probiotics rich in *Bifidobacterium* improve the response of PD-1/PDL-1 inhibitors [36].

## 7. Conclusions

The microbiota is actually considered an independent organ that is able to change following various external insults and variables. The increasing incidence of tumor diseases and improved therapeutic approach to them, with better overall survival of cancer patients, has opened the way to the study of factors that could interfere positively and negatively on the effectiveness of the same anticancer treatments. In the age of chemo-free therapies, represented by more selective and less toxic compounds, we focus on strategies to reduce the incidence of events capable of altering the prognosis and response.

Although several data have been found from studies on microbiota changes during ICIs therapies, no report is actually present in the literature on the CAR T-cells therapy front, due to the short time from their approval and probably due to pollution in the intestinal environment caused by the numerous chemotherapies previously received from this particular setting of patients. Finally, although it appears that the most modern immunotherapy does not impact on the composition of the gut microbiota, controlled studies are still required to confirm this hypothesis.

## Figures and Tables

**Figure 1 ijms-21-08902-f001:**
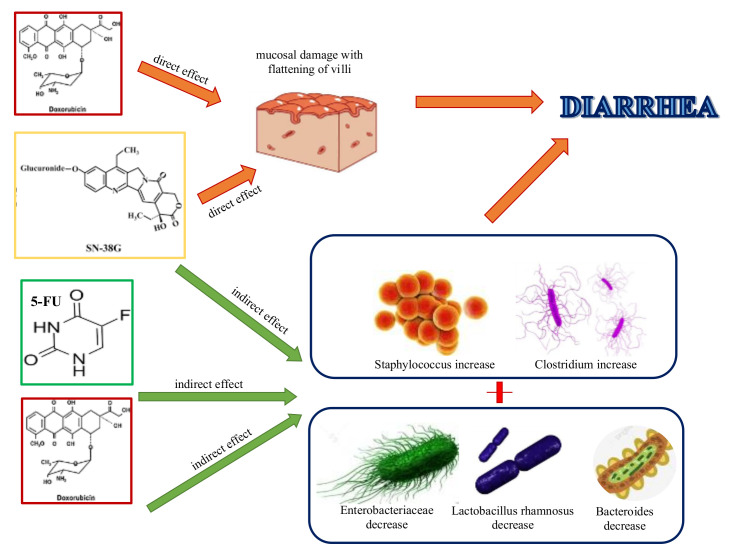
Conventional chemotherapies cause diarrhea by direct damage to the intestinal mucosa, flattening the villi (doxorubicin and irinotecan), and altering gut microbiota composition, by encouraging the increase of certain bacterial species, such as *Staphylococcus* and *Clostridium* spp., and decrease of others, like Enterobacteriaceae, *Lactobacillus*, and *Bacteroides*, followed by mucosal damage and diarrhea.

**Figure 2 ijms-21-08902-f002:**
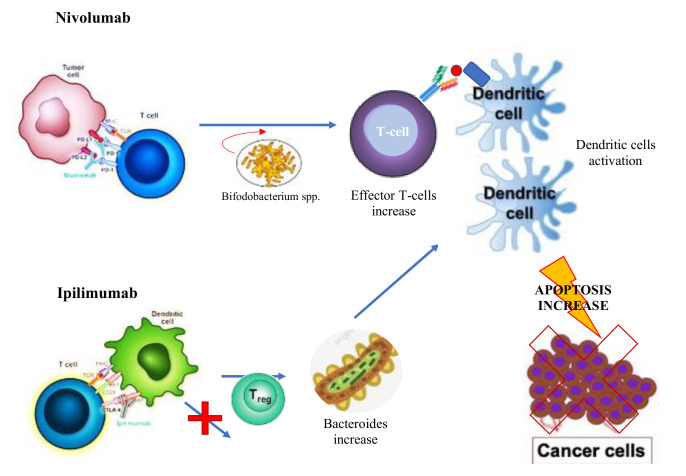
Anti-CTLA-4 immunotherapy (Ipilimumab) and Anti- PD-1 (Nivolumab) cause cancer cell apoptosis through their direct Tregs paralyzation, thus activating effector T cells and dendritic cells, also supported by the induced increase of certain bacterial species, such as *Bacteroidales*.

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
