# Peer review of "New Frontiers about the Role of Human Microbiota in Immunotherapy: The Immune Checkpoint Inhibitors and CAR T-Cell Therapy Era"

_ijms, 2020, doi:10.3390/ijms21238902_

Round 1
Reviewer 1 Report
The authors put together a review on the interplay between the microbiome and cancer immunotherapy, a very timely and relevant topic. The whole manuscript requires improvement as language is concerned, from grammar to style (some examples given below).
Specific comments
Title
“Car” -> CAR. CAR is an acronym standing for Chimeric Antigen Receptor and as such it’ always all capital letters
Abstract
“If we have obtained several data…” -> While we have obtained abundant data
Cytotoxic T lymphocyte-associated protein 4 is improved -> add the acronym CTLA-4
“Car T-cells” -> CAR T cells, plural of CAR T-cell
“Car T-cells seem to be 24 not interfering with microbiota, although the numerous previous therapies may have caused 25 permanent damage, by infusing the same reliability of the data we might have obtained.” – this sentence makes no sense
Introduction
“Several evidences” -> Several lines of evidence
“Is a time-dated acquisition that the cancer cells proliferation is a consequence of failure of
38 endogenous immunological control “ – what’s a “time-dated acquisition”? And a sentence shouldn’t start with “Is” unless it’s a question
line 78: “overexpress” -> overexpressed
line 101: “die for apoptosis” -> die via apoptosis
“are like “immortalized” in a stable cell-cycle arrest” – “are like” is too colloquial
line 408: “Actually” too colloquial, just remove word, it adds no meaning anyway
Still don’t understand why sometimes it’s “CAR T cells” (correct way) and sometimes it’s “Car T cells” (incorrect way)
Line 450: do thee clinical trials have clinicaltrials.gov numbers? If so, these should be indicated
Author Response
Dear referee,
thanks for your good evaluation of my manuscript.
I made the following changes, as you suggested:
In the Title:
- I modified "Car" with "CAR" in the title and all manuscript;
In the Abstract:
- I changed "if" with "While we have obtained abundant data";
- I added the acronym "CTLA-4" to Cytotoxic T lymphocyte-associated protein 4;
- I preferred CAR T-cell therapy;
- I changed the second to last sentence in “CAR T-cell therapy seem to be not interfering with microbiota, although the numerous previous therapies may have caused permanent damage, by obscuring the data we might have obtained."
In the Introduction:
- I changed “Several evidences” with "Several lines of evidence";
- I modified "Is a time dated acquisition that cancer cells..." in "Old acquisitions reported that cancer cells...";
- Line 78: I changed "overexposes" with "overexpressed";
- Line 101: “die for apoptosis” was replaced with "die via apoptosis";
- Line 104: I deleted "like" leaving “are immortalized”;
- Line 408: I deleted “Actually”;
- Line 450: I added clinicaltrials.gov numbers, as requested.
Finally, I have done a good deal of language throughout the manuscript, to make it more fluent, with particular attention to the use of locutions, appropriate tense and words.
I hope the changes I made will be to your liking and I give you our best regards.
Vanessa Innao
Reviewer 2 Report
A review article entitled, "New frontiers about the role of human microbiota in response to immunotherapy: the immune checkpoint inhibitors and CAR-T cell therapy era", written by Innao, et al. summarizes the latest understanding about microbiome effects on the effectiveness and toxicity of conventional chemotherapies, ICIs and CAR-T immunotherapies. The review is well organized with proper subsections to cover the latest advances in this field. The reference papers cited in the review are well-reasoned and pertinent. I wholeheartedly recommend its publication, but I must point out that there are quite a few infelicities in the English used in the text, which may adversely affect the readability of this article. As an example, in the Abstract:
1) Even though “microbiota” can be used as either a singular or plural noun, it appears to be used predominantly in the plural in this paper. A definitive choice should probably be made.
2) Is it “Car-T cell” or “Car T-cell”?
“subordinated to their ability to not stop treatment”. I suggest “subject to” rather than “subordinated to”.
“we have less data about alterations in intestinal flora in those ongoing to immunotherapy”. I suggest “undergoing” or “continuing” according to the meaning you intend, rather than “ongoing to”.
“Car T-cells seem to be not interfering with microbiota”. I suggest using “CAR T-cell therapy” [or Car-T cells; see above] for entire paper and the title.
“by infusing the same reliability of the data we might have obtained”. Here I suggest “by obscuring the data we might have obtained”, although I cannot be entirely certain of the meaning you intend.
3) Furthermore, the title seems to need some work as well.
Author Response
Dear referee,
thanks for your good evaluation of my manuscript.
I have done a good deal of language throughout the manuscript, to make it more fluent, paying a particular attention to the use of locutions, appropriate tense and words, as you suggested.
In particular, as regards your comments, I made the following changes:
- I chose to use "microbiota" always as singular, correcting the tense if wrong, like in line 201;
- I modified “Car-T cell” in "CAR T-cell" throughout the manuscript;
- In the Abstract I changed “subordinated to" with "subject to";
- In the sentence “we have less data about alterations in intestinal flora in those ongoing to immunotherapy”, I changed "ongoing to" with “undergoing”;
- I modified “by infusing the same reliability of the data we might have obtained” with “by obscuring the data we might have obtained”.
- Finally, I shortened the title by eliminating “in response to”.
I hope my changes have your approval.
My best regards.
Vanessa Innao